## RESEARCH ARTICLE

# The short isoform of Tango1 is dispensable for zebrafish survival but is required for skeletal patterning and integrity

Elizabeth A. Lawrence[1], Maria Esther Prada-Sanchez[1], Qiao Tong[1], Bianca Fernandes[1], Rebecca M. Chatwin[1], Michael Donohue[2], Brian Link[2], David J. Stephens[1,*] and Chrissy L. Hammond[1,‡]

## ABSTRACT

Collagen is the most abundant protein in the human body, providing structural stability to connective tissues. It organises and interacts with other proteins to form a complex extracellular matrix (ECM), with loss of collagen in the ECM seen in diseases such as osteoarthritis and osteoporosis. As collagen, and other ECM components, are atypically large proteins, they require specific endoplasmic reticulum (ER) export machinery. A key player in the export of procollagen from the ER is the *MIA3* gene product, TANGO1. We introduced mutations to both *tango1* isoforms in zebrafish independently to understand the importance of the previously unexplored short isoform in zebrafish development and tissue homeostasis. We show that the long isoform of *tango1* (*tango1L*) is mostly able to compensate for loss of the short isoform (*tango1S*) in larvae. However, non-collagenous components of the ECM (such as proteoglycans) were disrupted during development, leading to abnormal matrix patterning, visible by electron microscopy. Adult *tango1S* zebrafish show altered spinal morphology and changes to intervertebral discs, suggesting that *tango1S* plays a role in skeletal patterning and homeostasis that is independent of the long isoform.

KEY WORDS: TANGO1, Extracellular matrix, Zebrafish, Collagen secretion, Matrix secretion

## INTRODUCTION

The extracellular matrix (ECM) is a complex network of proteins that provides tissues with structural integrity as well as an environment conducive to cell signalling, proliferation and differentiation. The ECM consists of over 150 protein components ((Naba et al., 2012, 2017; Beachley et al., 2015; Adams, 2018), with the most abundant family being the collagens, which account for around 30% of the dry weight of humans (Deshmukh et al., 2016). Collagens confer different biomechanical properties through their assembly, such as elasticity and resistance to mechanical load in ligaments and provision of tensile strength in bone. Collagens also have an important role in organising the matrix

[1]School of Biochemistry and Molecular Medicine, Biomedical Sciences Building, University of Bristol, Bristol, BS8 1TD, UK. [2]Department of Cell Biology, Neurobiology and Anatomy, Medical College of Wisconsin, Milwaukee, WI 53226, USA.
*Deceased

‡Author for correspondence (chrissy.hammond@bristol.ac.uk)

E.A.L., 0000-0002-8953-7183; M.E.P.-S., 0000-0003-0118-7586; B.L., 0000-0002-7173-2642; D.J.S., 0000-0001-5297-3240; C.L.H., 0000-0002-4935-6724

through their interactions with other ECM components such as proteoglycans (Mayne, 1989) and elastin (Wise and Weiss, 2009). Mutations in collagen genes prevent the formation of a fully functional ECM and can result in conditions such as osteogenesis imperfecta (Cole and Dalgleish, 1995; Willing et al., 1996), Ehlers-Danlos syndrome (Nuytinck et al., 2000) and Stickler syndrome with associated premature osteoarthritis (Vikkula et al., 1995; Richards et al., 1996; Couchouron and Masson, 2011; Bonafe et al., 2015). Degradation of collagen in the ECM can also result in chronic conditions such as osteoporosis (Shuster, 2020) and osteoarthritis. During the onset and progression of osteoarthritis, the ECM secreted by chondrocytes changes with aggrecan content of the matrix decreased and type I collagen increased (Sandell and Aigner, 2001; Pearle et al, 2005; Martel-Pelletier et al., 2008; Lahm et al., 2010).

Components of the ECM, including collagens, are translated at the endoplasmic reticulum (ER) before travelling through the early secretory pathway to the Golgi apparatus, where they undergo posttranslational modifications and are directed for secretion from the cell (Hellicar et al., 2022). The effective transport of these proteins from the ER, through the ER-Golgi intermediate compartment (ERGIC) to the Golgi in the early secretory pathway is critical for the integrity of the ECM that is formed (Canty and Kadler, 2005). The first step of this process is driven by the assembly of the COPII complex (Barlowe et al., 1994) at the ER membrane. The formation of the COPII coat is initiated by activation of Sar1 by the ER-associated guanine nucleotide exchange factor Sec12. GTP binding to cytoplasmic Sar1 triggers a conformational change and association of Sar1 with the ER membrane (Nakano and Muramatsu, 1989; Barlowe et al., 1993; Kuge et al., 1994). Sar1 recruits the inner coat components Sec23/24 before the outer coat Sec13/Sec31 heterotetramers are recruited (Matsuoka et al., 1998; Bi et al., 2002). These COPII proteins assemble at relatively stable sites on the ER membrane, termed transitional ER (Orci et al., 1994) where COPII-coated vesicles with a diameter of 60-80 nm can bud and combine to form the ERGIC (Schweizer et al., 1988). Together, these structures form ER exit sites (ERES) (Hughes et al., 2009).

A range of proteins have been identified that are involved in organising and regulating ERES membrane dynamics and recruitment of cargo to these sites, including TANGO1, which is encoded by the *MIA3* gene and present as both a long (TANGO1L) and short (TANGO1S) isoform. TANGO1 has been shown to be a critical component of the early secretory pathway, with both isoforms being shown to have a range of complementary and divergent functions in cells (Raote et al., 2018; McCaughey et al., 2021; Raote and Malhotra, 2021; Arnolds and Stoll, 2023). TANGO1S is a splice isoform of TANGO1L, with 785 amino acids that encode the cytoplasmic domains arising from the same exons as those in TANGO1L. The cytoplasmic domains of

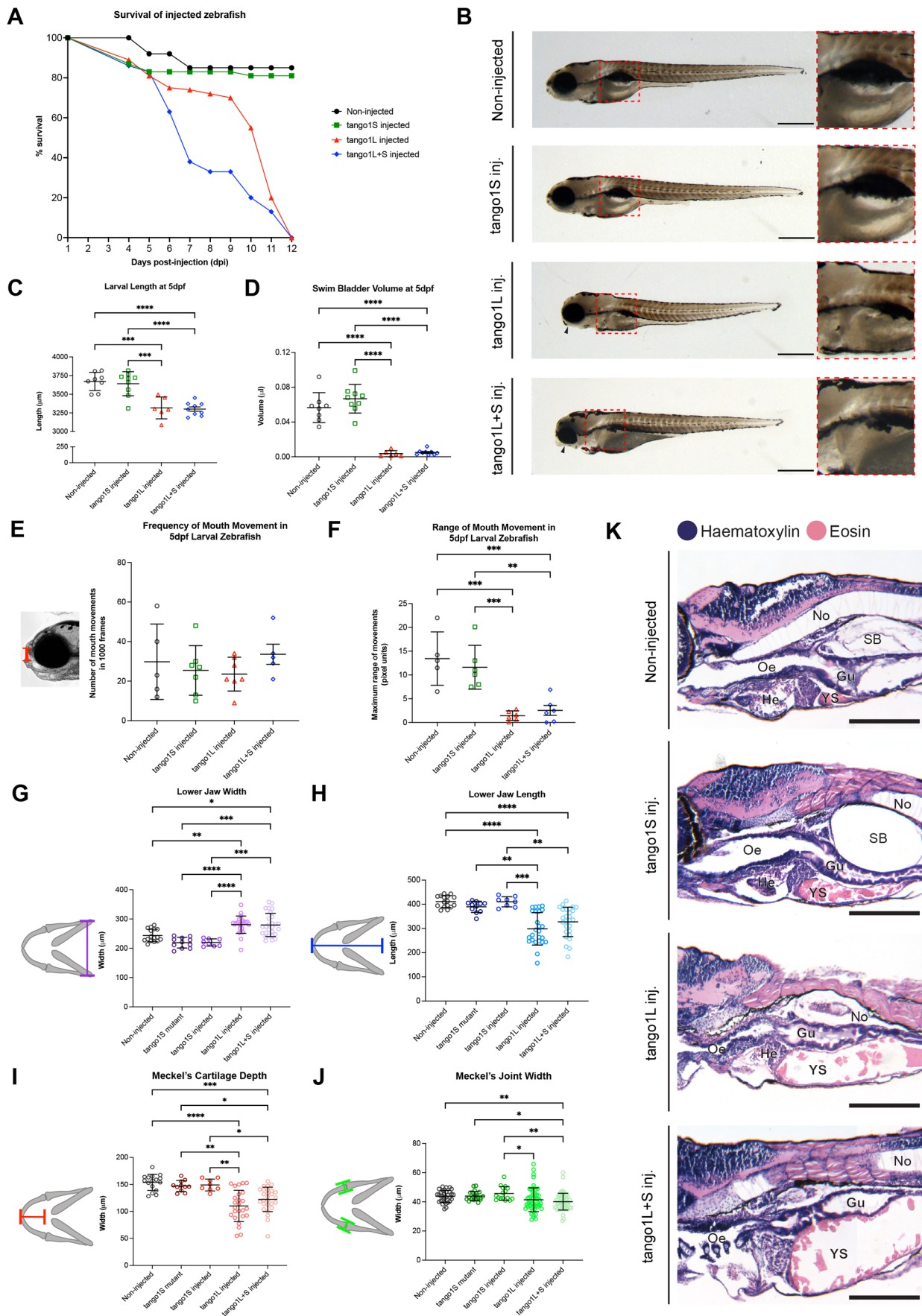

**Fig. 1.** See next page for legend.

**Fig. 1. Loss of Tango1L causes general development changes in larval fish consistent with delayed development and severe morphological abnormalities to the lower jaw of developing fish, which prevents feeding.** (A) Survival rates of injected zebrafish compared to non-injected control fish. For each experimental group, the survival of 70 larvae per group was tracked. (B) Representative brightfield lateral-view images of 5 dpf zebrafish from control or injected groups; inset image shows swim bladder region in more detail, with location of measurements taken to calculate swim bladder volume annotated on control fish. Scale bars: 500 µm. (C,D) Quantification of larval length (C) and swim bladder volume (D) from 5 dpf larval zebrafish, n=8, 9, 6 and 9 for non-injected controls, *tango1S*, *tango1L* and *tango1L+S* crispants, respectively. (E,F) Quantification of number of mouth movements (E) and range of mouth movement (F). Location of mouth movement measurements taken annotated by the red line on image to the left of graph in E; n=4,7, 7, and 3 for non-injected controls, *tango1S*, *tango1L* and *tango1L+S* crispants, respectively. (G-J) Quantification of lower jaw width (G), length (H), Meckel's cartilage depth (I), and Meckel's joint width (J) from 5 dpf fish. Schematics showing the location of measurements taken are shown to the left of each graph; n=16, 11, 8, 25 and 27 for non-injected controls, *tango1S* mutants, *tango1S* crispants, *tango1L* crispants and *tango1L+S* crispants, respectively. (K) Representative brightfield images of H&E-stained 5 dpf zebrafish larvae sectioned laterally. No=notochord, Oe=oesophagus, He=heart, YS=yolk sack, Gu=gut, SB=swim bladder. Scale bars: 100 µm. One-way ANOVA with Tukey's multiple comparisons test performed in all graphs. Data are mean with s.e.m. *$P \leq 0.05$, **$P \leq 0.01$, ***$P \leq 0.001$, ****$P \leq 0.0001$.

TANGO1 include a proline-rich domain (PRD), two coiled-coil domains (CC1 and CC2) and a tether for the ERGIC at the ER (TEER) domain. CC2 mediates the interaction with cTAGE5 (Lekszas et al., 2020; Saito et al., 2011), another protein found at ERES. Within CC1 is the TEER domain, which helps recruit ERGIC membranes to the ERES, providing the necessary membrane for forming large transport carriers. The C-terminal proline-rich domain (PRD), located in the cytoplasm, is crucial for TANGO1's localisation to ERES. The PRD contains repetitive motifs that interact with the inner COPII coat components, specifically Sec23/Sec24 (Ma and Goldberg, 2016), and recruits Sec16 (Maede et al., 2016). By binding these factors, the PRD plays a critical role in organising the COPII machinery and coordinating the assembly of the export platform at the ERES (Ma and Goldberg, 2016; Maeda et al., 2017). In addition to these cytoplasmic domains, the 1907 amino acid TANGO1L isoform has an ER-luminal SH3 domain that resides within the ER lumen and is primarily responsible for recognising and binding cargo (Ishikawa et al., 2016; McCaughey et al., 2021). While it can directly bind certain collagens like type IV (Arnolds and Stoll, 2023), its main interaction in vertebrates is with the collagen chaperone HSP47, forming a complex that facilitates the export of various collagen types (Ishikawa et al., 2016). At the N-terminal, TANGO1S has only 15 amino acids which sit within the ER lumen and is therefore thought to lack the ability to directly bind cargoes (Raote et al., 2018).

In humans, complete loss of TANGO1 is embryonically lethal due to an absence of bone mineralisation (Guillemyn et al., 2021) and point mutations in TANGO1 that result in exon skipping leads to a range of complex skeletal abnormalities (including short stature, osteopenia, and platyspondyly), insulin-dependent diabetes mellitus and sensorineural hearing loss (Cauwels et al., 2005; Lekszas et al., 2020). In cells with the corresponding protein truncation, an inability of TANGO1 to localise to ERES and a reduction in type I collagen secretion is seen (Cauwels et al., 2005; Lekszas et al., 2020). Intracellular retention of type I procollagen and a reduction in type I collagen matrix is also seen in TANGO1-knockout (KO) human cells (McCaughey et al., 2021), with the

changes seen dependent on which isoforms are affected, with the most severe consequences resulting from loss of both the long and short isoforms. This includes significant disruptions to the organisation of the ER–Golgi interface, causing dramatic changes in ultrastructure visible by electron microscopy; changes to gene and protein expression of key early secretory pathway components; and an induction of ER stress in cells lacking both isoforms (McCaughey et al., 2021). Crucially, the loss of both isoforms in cell models impairs the secretion of a broad range of secretory cargo, encompassing not only large proteins such as procollagen but also small soluble proteins (McCaughey et al., 2021; Raote et al., 2018). The core cytoplasmic domain, present in both isoforms, appears essential for the fundamental role of TANGO1 in the early secretory pathway as the severe phenotypes observed in cells lacking both isoforms can be largely rescued by reintroducing either recombinant TANGO1S or TANGO1L (McCaughey et al., 2021).

Studies in zebrafish have shown that loss of Tango1 function through treatment with peptide inhibitors or CRISPR/Cas9-mediated deletion causes a reduction in secretion of several ECM components and altered tissue architecture in developing fish (Clark and Link, 2021; Raote et al., 2024). Zebrafish with a large deletion in *tango1L*, which does not affect Tango1S, show protein retention in the ER, resulting in an upregulation of ER-stress pathways and aberrant ECM secretion, including an inability of chondrocytes to secrete type II collagen; severe craniofacial malformations; and a failure to survive to adulthood (Clark and Link, 2021). The phenotype of these fish has similarities to the *feelgood* (Melville et al., 2011) and *crusher* (Lang et al., 2006) zebrafish lines, which have mutations affecting COPII coat proteins. Similarly, medaka have been used to show the necessity of the Tango1L isoform for survival and type II collagen export in some tissues (Yasuda et al., 2025).

Although previous work in zebrafish and medaka has demonstrated an important role for Tango1 and its binding partner Ctage5 in secretion of large ECM proteins (particularly collagens), this work has largely focused on the long isoform, therefore questions remain about the relative importance and function of the short isoform. Here, we show that Tango1L largely compensates for the loss of Tango1S; however, there are some functions that appear to be dependent on the short isoform, including secretion of non-collagenous ECM components (such as proteoglycans) during development, which contribute to skeletal abnormalities in adult zebrafish. These skeletal abnormalities resemble the phenotype of humans with a single base pair mutation in TANGO (Lekszas et al., 2020). Our findings in zebrafish lacking Tango1L align with those previously published by the Link lab (Clark and Link, 2021) and further show that loss of both Tango1 isoforms leads to a more severely disrupted ECM phenotype in larval zebrafish, further supporting a role for the short isoform in ECM secretion and homeostasis.

## RESULTS

### Both Tango1 isoforms have a role in the early stages of zebrafish development, but only the long isoform is essential for survival

Following on from work by the Link lab (Clark and Link, 2021), we sought to determine whether mutation of either Tango1 isoform was detrimental to survival. Following injection with guide RNAs targeting: the short isoform (Tango1S), the long isoform (Tango1L) or both isoforms (Tango1L+S) (Fig. S1), we recorded the survival of larvae for 12 days post fertilisation (dpf). Survival of injected larvae was compared to that of both non-injected and sham-injected

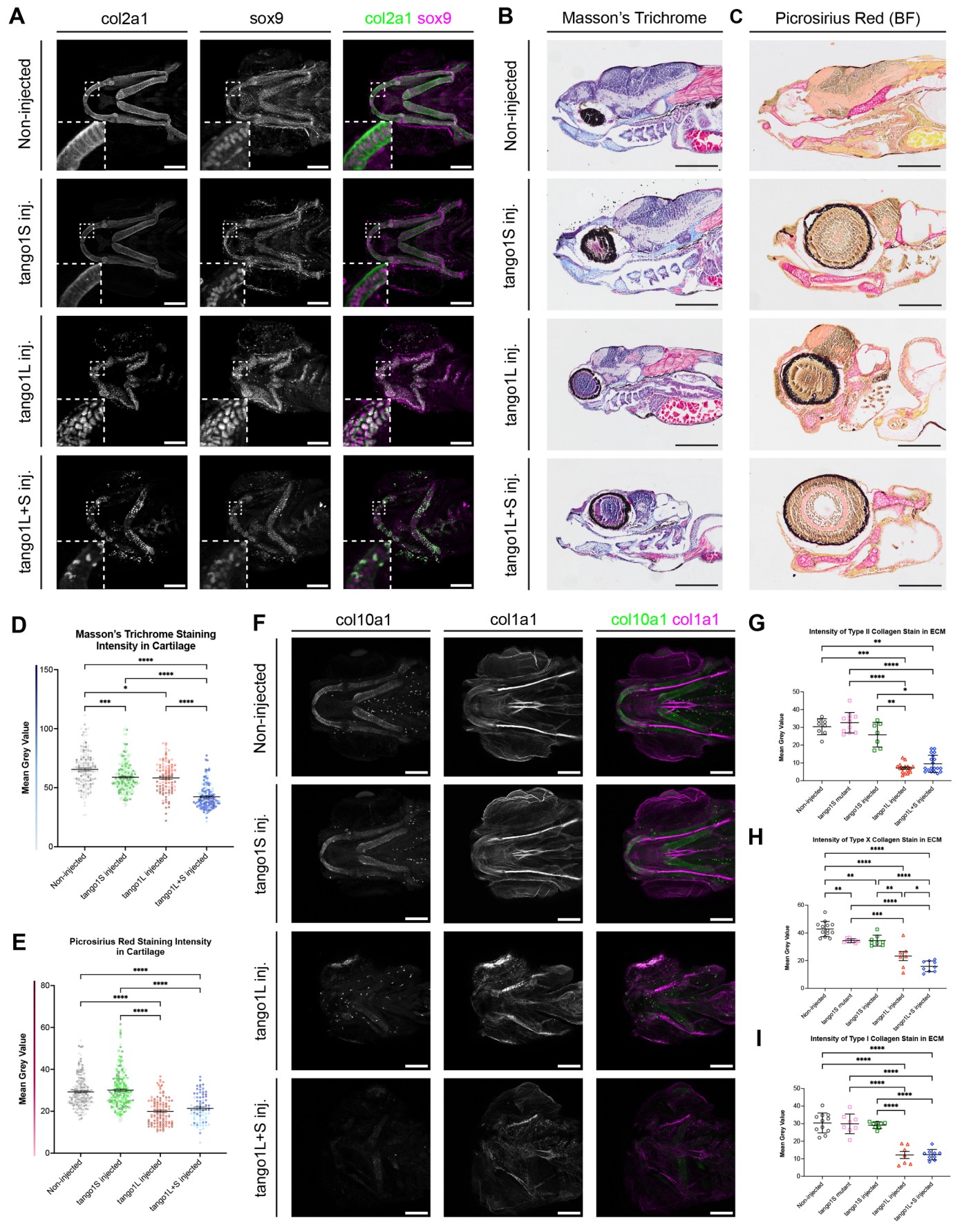

**Fig. 2.** See next page for legend.

**Fig. 2. Collagen deposition appears most disrupted in the lower jaw cartilage of larval zebrafish lacking both Tango1 isoforms.**
(A) Representative maximum projections of ventral view confocal image stacks from 5 dpf zebrafish immunostained for type II collagen and sox9. Scale bars: 100 µm. (B,C) Masson's trichrome (B) and Picrosirius Red (C)-stained lateral sections of 5 dpf zebrafish. Scale bars: 250 µm. (D,E) Quantification of Masson's trichrome (D) and Picrosirius Red (E) staining intensity in the ECM from brightfield images of injected fish, *n*=3 fish for all groups (individual points represent the intensity value from one section). (F) Representative maximum projections of ventral view confocal image stacks from 5 dpf zebrafish immunostained for type X and type I collagen. Scale bars: 100 µm. (G-I) Quantification of type II collagen (G), type X collagen (H), and type I collagen (I) staining intensity in the lower jaw cartilage ECM of 5 dpf zebrafish larvae. In G, *n*=8, 11, 7, 16, 19; H, *n*=13, 9, 8, 7, 9; and I, *n*=11, 8, 8, 7, 9 for non-injected controls, *tango1S* mutants, *tango1S* crispants, *tango1L* crispants and *tango1L+S* crispants, respectively. Kruskal–Wallis test with Dunn's multiple comparisons test performed in C,D and F. One-way ANOVA with Dunn's multiple comparison test performed in G and I. Data are mean with s.e.m.. *$P \leq 0.05$, **$P \leq 0.01$, ***$P \leq 0.001$, ****$P \leq 0.0001$.

controls (embryos injected with Cas9 but no guides). No change was seen in the survival or general development phenotype of non-injected versus sham-injected fish, so to reduce preserve reagents and reduce the number of embryos subjected to injections, we proceeded to use non-injected controls for the remaining experiments. Larvae with a deletion in *tango1L* or *tango1L+S* show a significant reduction in survival from 6 dpf, with no larvae from either group surviving past 12 dpf (Fig. 1A). Zebrafish with a double isoform deletion show a dramatic drop in survival earlier than those with just a *tango1L* deletion, with over 50% of *tango1L+S* fish dead by 7 dpf compared to 11 dpf in *tango1L* fish (Fig. 1A). The general development of fish with a *tango1L* or *tango1L+S* deletion also appeared to progress more slowly, with these fish showing a significant decrease in larval length and swim bladder volume at 5 dpf (Fig. 1B-D), and a reduction in lower jaw protrusion, as previously shown in Clark and Link, 2021 (black arrowheads in Fig. 1B). Larvae with a mutation in *tango1S* show survival rates comparable to those of non-injected controls (Fig. 1A), and these fish survive to adulthood; at 18 months old, these fish are viable and fertile.

As the drop off in survival of *tango1L* and *tango1L+S* fish coincides with the onset of independent feeding (Strähle et al., 2012) and due to the observed jaw phenotype here and in Clark and Link, 2021, we measured the number and range of jaw movements for larval zebrafish at 5 dpf to ascertain whether these fish were capable of feeding. This showed that although the number of movements made was comparable across groups (Fig. 1E), the range of these mouth movements was significantly decreased in *tango1L* and *tango1L+S* larvae (Fig. 1F). Analysis of lower jaw morphology revealed that *tango1L* and *tango1L+S* fish had drastically different jaw morphologies at 5 dpf across all parameters measured (Fig. 1G-J). Histological analysis revealed no apparent changes to the digestive system in these fish (Fig. 1K). General development indicators and lower jaw morphology were affected to a similar extent whether *tango1S* was mutated (*tango1L+S*) or not (*tangoL*). As fish injected with Tango1S guides survive to adulthood, we were able to generate a stable mutant line with a 2 bp insertion in exon 1, resulting in a premature stop codon 8 amino acids downstream of the insertion site (Fig. S1). These fish are referred to as *tango1S* mutants throughout the rest of this paper, with the injected crispants referred to as *tango1S* crispants. No significant changes to lower jaw morphology was seen between wild-type fish, *tango1S* mutants or crispants (Fig. 1G-J).

## The amount of collagen present in the cartilage ECM is decreased in zebrafish with a mutation in *tango1L*, and to a less severe degree in those with a *tango1S* mutation

As TANGO1 acts as a procollagen receptor (Saito et al., 2009; Ishikawa et al., 2016) and zebrafish lacking the long isoform have been previously shown to accumulate intracellular type II collagen in the lower jaw cartilage (Clark and Link, 2021), we next examined whether the short isoform has a role in collagen secretion. Using wholemount immunostaining for type II collagen, we saw an accumulation within chondrocytes (overlapping with intracellular sox9 labelling) and a lack of type II collagen in the surrounding lower jaw ECM of *tango1L* and *tango1L+S* fish (Fig. 2A). This supports the previous findings by the Link lab that type II collagen becomes trapped inside the ER of 4 dpf zebrafish with a deletion in *tango1L* (Clark and Link, 2021). The expression of type II collagen, visualised using the *tg(col2a1aBAC:mCherry); (Ola.Sp7:NLS-GFP)* double transgenic reporter fish, was unchanged in *tango1L* and *tango1L+S* fish (Fig. S2), demonstrating that loss of tango does not impair chondrocyte differentiation. No change to protein distribution was seen in *tango1S* mutants or *tango1S* crispants, with levels of type II collagen throughout the ECM surrounding the lower jaw chondrocytes comparable to controls (Fig. 2A). The reduction of type II collagen protein in the lower jaw ECM was quantified and found to be significantly decreased in *tango1L* and *tango1L+S* crispants compared to non-injected controls, whereas no difference was seen in the amount of type II collagen in the lower jaw cartilage ECM in *tango1S* mutants or *tango1S* crispants (Fig. 2G). The formation of a collagen-rich ECM by *tango1S* crispants was further confirmed by histological staining with Masson's trichrome (Fig. 2B) and Picrosirius Red (Fig. 2C). No significant change in collagen staining intensity was seen between non-injected controls and *tango1S* crispants stained with Picrosirius Red (Fig. 2E); however, a decrease in the intensity of Masson's trichrome staining in the cartilage elements of *tango1S* crispants was observed (Fig. 2D). Given the similarities between *tango1S* mutants and *tango1S* crispants in other areas measured, we predicted that the stable mutants would have a comparable decrease in Masson's trichrome staining intensity. As trichrome stains total collagen, rather than a specific collagen protein, this suggests changes to secretion of minor collagens in these fish. Both *tango1L* and *tango1L+S* fish showed a decrease in the mean staining intensity of both Picrosirius Red and Masson's trichrome when compared to that in non-injected wild-type fish and *tango1S* fish (Fig. 2D,E). In zebrafish with a double isoform deletion, the staining intensity of Masson's trichrome in the cartilage ECM was significantly lower than that in fish with a deletion in *tango1L* (Fig. 2D). Again, this indicates that the short isoform may have a partially compensatory role when the long isoform is lost.

To further understand whether minor collagens in the lower jaw cartilage ECM were affected in a similar way to type II upon loss of Tango1, we used immunostaining to visualise two minor collagens: non-fibrillar type X and fibrillar type I collagen (Fig. 2F). This revealed a decrease in both collagens in the lower jaw cartilage ECM of *tango1L* and *tango1L+S* fish (Fig. 2H,I). In *tango1S* mutants and *tango1S* crispants, no change in type I collagen staining was measured, but a decrease in type X collagen staining in the cartilage ECM was seen (Fig. 2H), although there was still significantly more of this collagen present than in fish lacking the long isoform.

## Non-collagenous components of the cartilage ECM are affected by mutation of *tango1* in larval zebrafish

We next sought to determine whether secretion of other ECM components was affected by deletion of different Tango1 isoforms.

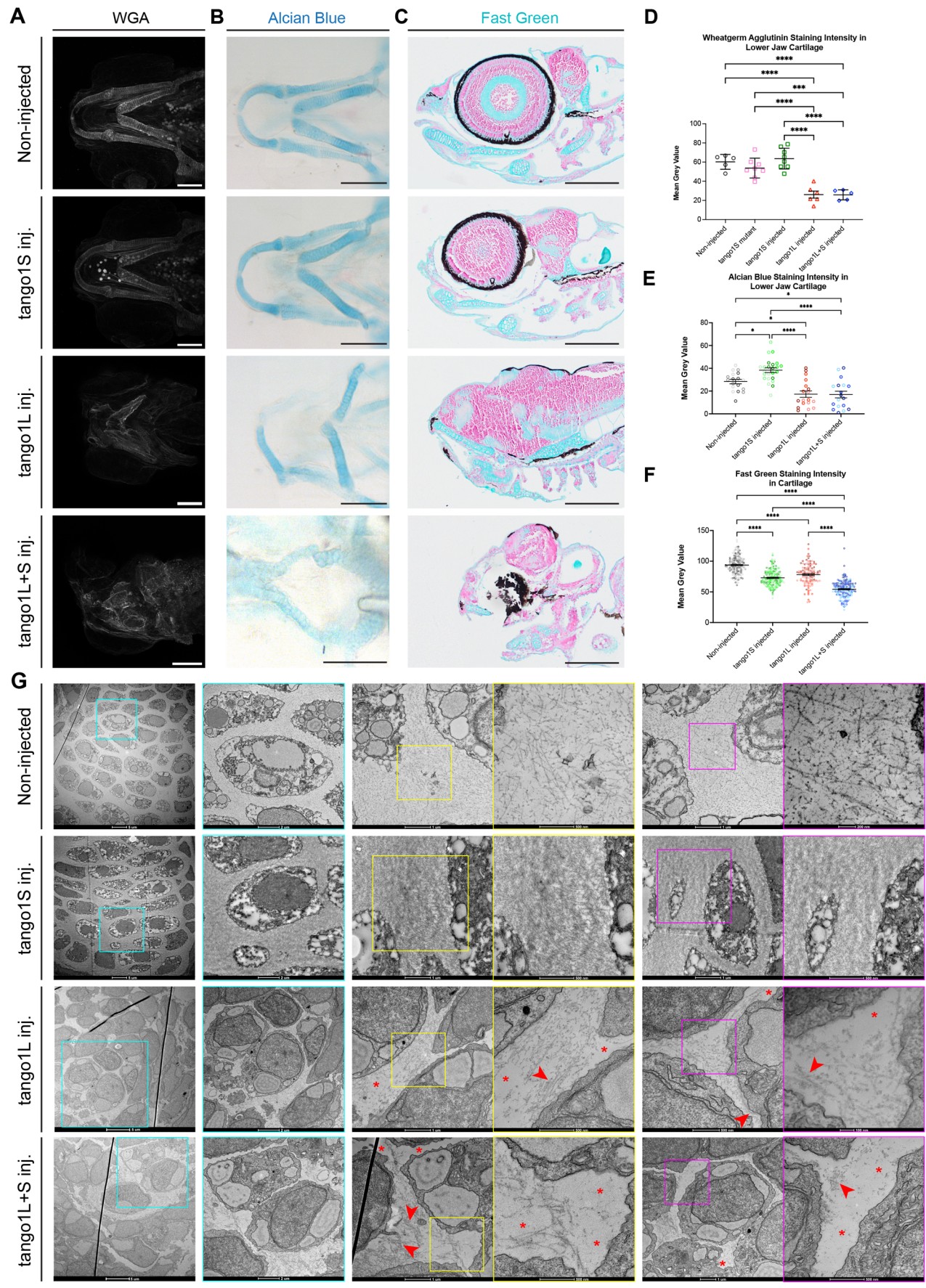

**Fig. 3.** See next page for legend.

**Fig. 3. Deposition of other cartilage ECM components is affected in fish lacking the long and/or short isoform of Tango1.** (A) Representative maximum projections of ventral view confocal image stacks from 5 dpf zebrafish stained with WGA. Scale bars: 100 μm. (B,C) Representative brightfield images of whole 5 dpf larvae stained with Alcian Blue (B) or lateral sections stained with Fast Green (C). Scale bars: 250 μm. (D-F) Quantification of WGA (D), Alcian Blue (E), and Fast Green (F) staining intensity in the lower jaw cartilage ECM. In D, *n*=5, 8, 8, 6, 5 (for non-injected controls, *tango1S* mutants, *tango1S* crispants, *tango1L* crispants and *tango1L+S* crispants, respectively); E, *n*=3, 4, 3 and 3 (for non-injected controls, *tango1S*, *tango1L* and *tango1L+S* crispants, respectively); F, *n*=3 for all groups. (G) Electron micrographs of cartilage in 5 dpf zebrafish larvae. Location of higher-resolution images shown by coloured boxes in left-hand image. One-way ANOVA with Dunn's multiple comparison test performed in D, Kruskal–Wallis test with Dunn's multiple comparison test performed in E and F. Data are mean with s.e.m. *$P \leq 0.05$, ****$P \leq 0.0001$.

We stained larvae with wheatgerm agglutinin (WGA) to visualise the carbohydrate component of the ECM. This dye binds to N-acetyl-glucosamine (GlcNAc) and n-acetyl-neuraminic acid (NANA) in proteoglycans with antibody-like affinity (Ohno et al, 1986). No change was seen to staining intensity or distribution between non-injected controls, *tango1S* mutants or *tango1S* crispants; however, WGA staining in *tango1L* and *tango1L+S* fish was dramatically reduced in the cartilage ECM (Fig. 3A,D). This reduction in WGA staining was mirrored by a reduction in staining intensity of alcian blue (Fig. 3B,E) and Fast Green (Fig. 3C,F) suggesting a reduction in proteoglycan content of the cartilage ECM in fish lacking the long isoform of Tango1. This reduction in proteoglycan content was most severe in zebrafish lacking both isoforms of Tango1 (Fig. 3F). Using transmission electron microscopy (TEM), we were able to visualise the ultrastructural organisation of the ECM in the lower jaw cartilage. In 5 dpf control larvae, matrix was evenly distributed between chondrocytes (Fig. 3G). In *tango1S* crispants, a denser matrix was visible between chondrocytes, with an abundance of fibrillar proteins present (Fig. 3G). By contrast, in larval fish injected with Tango1L guides, the matrix was unevenly deposited (Fig. 3G), with some areas showing a density of matrix fibres arranged in a predominantly perpendicular orientation (red arrowheads in Fig. 3G) and others showing an absence of visible matrix components (red asterisks in Fig. 3G). Although mutation of the short or long isoform of Tango1 caused a different matrix phenotype, both lines had a substantially disrupted cartilage ECM, indicating a change in the relative balance of proteins in the cartilage ECM causing matrix misassembly. In zebrafish with mutations in both isoforms of Tango1, a similar phenotype to Tango1L crispants was seen with some areas of dense matrix and others with very sparse matrix. The matrix disruption seen in the *tango1L+S* crispants appears more severe than that of the *tango1L* crispants with a more extreme difference between areas of denser matrix (red arrowheads in Fig. 3G) and those where matrix is almost completely lacking (red asterisks in Fig. 3G).

## ECM secretion defects are seen in other tissues of zebrafish with *tango1* deletions

Having seen extensive collagen secretion defects in the cartilage of fish with mutated *tango1*, we next wanted to investigate whether the secretion of major collagens in other tissue ECM was affected or whether this was a tissue-specific phenotype. We chose to look at the skin and muscle, which are functionally different and therefore have distinct ECM compositions. The main ECM component in skin is type I collagen, which we visualised using the fibrillar collagen fusion line *krt19:col1a1-GFP* transgenic (Morris et al.,

2018) and by immunohistochemistry. In non-injected controls and *tango1S* crispants, a regular lattice of type I collagen fibres can be seen along the trunk of the zebrafish larvae (Fig. 4A,B). Although this lattice of collagen fibres is visible in *tango1L* and *tango1L+S* fish, the organisation is altered and shows bright punctae of GFP visible along the trunk (Fig. 4A,C,D). These punctae were also visible in immunostained larvae (Fig. 4B) and appear to show type I collagen trapped within skin cells in a juxtanuclear position (Fig. 4B). These bright intracellular punctae are also seen in the *tango1S* mutants and *tango1S* crispants, but more rarely, with around 25% of fish showing punctae and at reduced frequency (typically less than ten per region visualised) (Fig. 4B,C,D). Electron microscopy of the zebrafish epidermis revealed that the ER was noticeably more electron dense and distended in epidermal cells of *tango1L* and *tango1L+S* crispants suggestive of an accumulation of protein in this organelle (Fig. 4E). In *tango1S* fish, the ER and Golgi compartments appeared more comparable to those in non-injected controls; however, in some cells the Golgi apparatus ultrastructure appeared to be disrupted with some regions showing fragmentation and others showing condensed stacks (Fig. 4E).

In addition to examining collagen rich tissues such as the cartilage and skin, we also wanted to compare the impact of *tango1* mutations on a less collagen rich-tissue: muscle. In cartilage and skin, collagen accounts for more than 60% of the dry weight of the tissue (Oikarinen, 1994; Cohen et al., 1998), whereas in skeletal muscle it accounts for less than 10% of the dry weight (Gillies and Lieber, 2011). To examine the muscle in our injected fish, we used immunostaining for skeletal myosin (a4.1025) to visualise muscle fibres in the jaw and trunk (Fig. 5A). Although no difference was seen between the number of somites, number of muscle fibres per somite or muscle fibre thickness in fish with a *tango1L* mutation, the packing of the fibres was disrupted (Fig. 5A-F). As this phenotype can reflect alterations to muscle attachments sites in the myotendinous junctions, which are collagen rich, requiring collagen XXII and collagen XII (Malbouyres et al., 2022), we next examined a marker for these sites, thrombospondin-4 (thsp-4). In zebrafish injected with Tango1L and L+S guides, thrombospondin-4 staining appeared punctate and uneven through the fish with brighter regions visible along the midline (Fig. 5G). The somite boundaries in these fish also appeared irregular, with the typical 'chevron' shape less clearly defined and almost completely lost in some regions of the trunk. In *tango1S* crispants, these somite 'chevrons' were still clearly visible; however, the junctions between somites were wider and less well-defined than in non-injected controls (Fig. 5G). These changes to myotendinous junctions were also visible using electron microscopy (Fig. 5H).

## Adult *tango1S* mutant zebrafish have altered spine morphology and signs of intervertebral disc degeneration

Having observed changes to the myotendinous junctions in Tango1S crispants during development (Fig. 5G), we used micro-computed tomography (μCT) scanning to visualise the spine of 14-month-old *tango1S* mutant zebrafish to ascertain whether these changes in development led to an adult phenotype. This showed that zebrafish with a *tango1S* mutation have a straighter spine (Fig. 6), reminiscent of the spinal flattening seen in humans with TANGO1 mutations (Lekszas et al., 2020). Vertebral misalignments were also visible in *tango1S* mutant fish from μCT scans (white arrowheads in Fig. 6B). Histological staining of the vertebral column also revealed the presence of fibrotic tissue and disorganised cellular morphology in the nucleus pulposus of *tango1S* mutants (black asterisks in Fig. 6B) as well as extension and 'bulging' of the annulus fibrosus in these fish

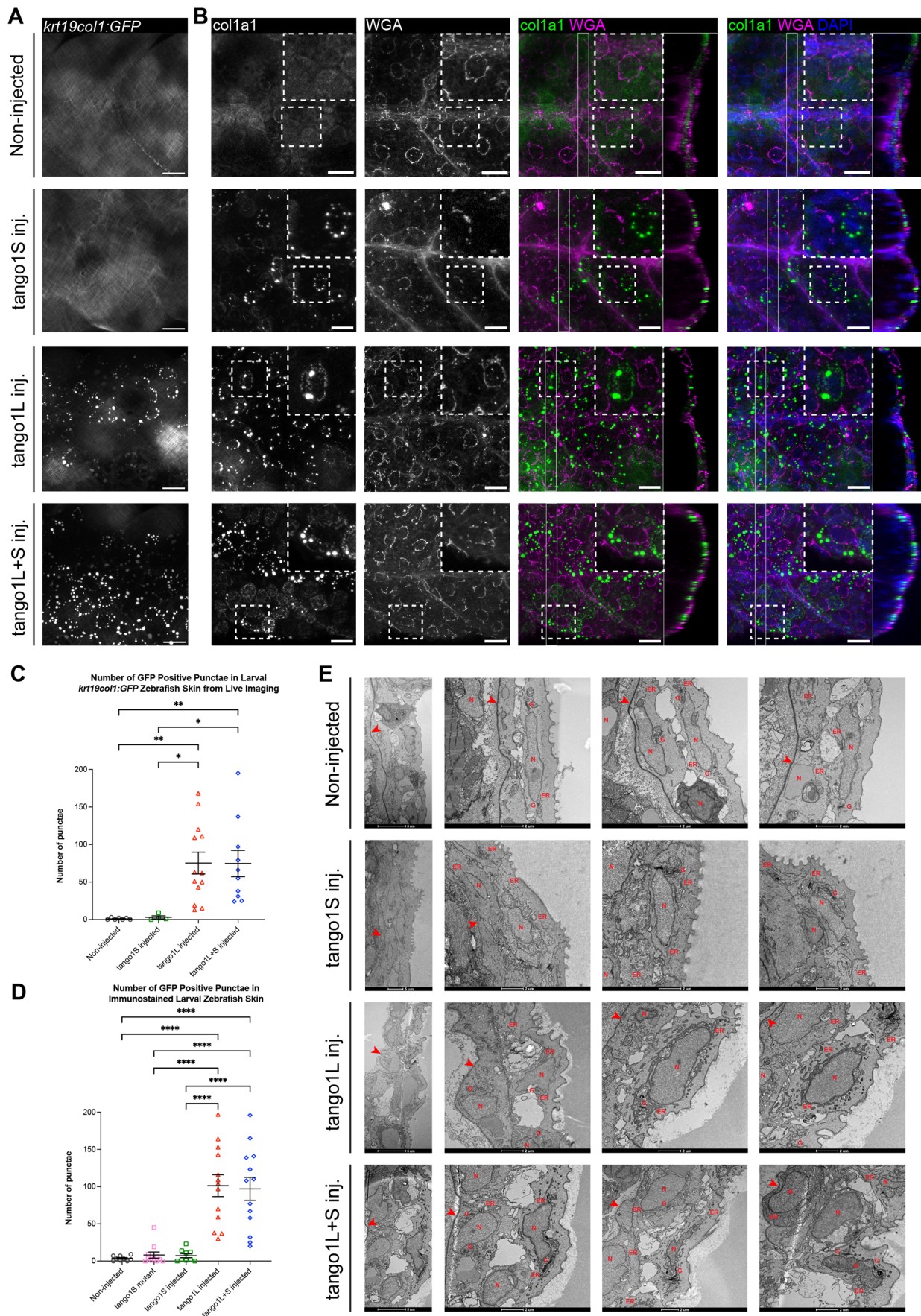

**Fig. 4.** See next page for legend.

**Fig. 4. Larval zebrafish lacking Tango1 show changes to collagen secretion in the skin.** (A,B) Representative maximum projections of lateral view confocal image stacks taken on the flank of 5 dpf Tg(krt19:col1a1-GFP) live zebrafish just above the cloaca (A) and after staining for GFP, WGA and DAPI (B). Extreme right of merged images shows a cross section through the skin (achieved by 3D projection of the confocal z stack and rotation to show cross section). Scale bars: 25 μm. (C,D) Quantification of number of GFP-positive punctae per field of view from live imaging and immunostained samples. In C, n=7, 4, 13, 10; D, n=9, 7, 13, 13; for non-injected controls, tango1S, tango1L and tango1L+S crispants, respectively. (E) Electron micrographs of 5 dpf zebrafish epidermis. Red arrowheads=basement membrane, N=nucleus, G=Golgi, ER=endoplasmic reticulum. Kruskal–Wallis test with Dunn's multiple comparison test performed in C and D. Data are mean with s.e.m. *$P \leq 0.05$, **$P \leq 0.01$, ****$P \leq 0.0001$.

(red asterisks in Fig. 6B). Interestingly, most of these changes, which have previously been associated with intervertebral disc degeneration in zebrafish (Kague et al., 2021), occur in the vertebral region around the last few ribs (pink boxes in Fig. 6), where the loss of spine curvature is most pronounced in tango1S mutant fish (Fig. 6D).

## DISCUSSION

It has previously been shown that zebrafish with a deletion in tango1L and Tango1L-KO medaka (Yasuda et al., 2025) fail to survive beyond the early larval stages (Clark and Link, 2021), but the importance of the tango1S isoform in survival and early zebrafish development has not been investigated. Studies in medaka suggest that Tango1S-KO fish hatch normally and can survive to 6 months post-hatching; however, the surviving population does not show the expected Mendelian ratio, suggesting a role for Tango1S that has yet to be reported (Yasuda et al., 2025). Here, we show that zebrafish with a mutation in exon 1 of tango1S have survival rates comparable to those of wild-type fish and reach adulthood. Zebrafish with a tango1L or tango1L+S deletion fail to survive past 12 dpf; however, fish with a deletion in both isoforms show a steep drop off in survival at an earlier stage than tango1L zebrafish. This supports the hypothesis that although only the long isoform is critical for zebrafish survival, the short isoform may have a role in essential developmental processes, allowing it to partially compensate for loss of the Tango1L isoform. This may enable Tango1S to slow the accumulated phenotype in these mutants and prolong survival.

One way in which the short isoform of Tango1 may help prolong survival in Tango1L fish compared to Tango1L+S fish is through facilitating secretion of cartilage ECM components to form a more functional ECM than can be secreted when both isoforms are deleted. The most abundant protein in the cartilage ECM is collagen, and tango1L has been shown to play a critical role in transporting procollagen out of the ER (Saito et al., 2009; Wilson et al., 2011; Nogueira et al., 2014; Ishikawa et al., 2016; Ma and Goldberg, 2016; Liu et al., 2017; Raote et al., 2017, 2018; Clark and Link, 2021) prior to its processing at the Golgi and eventual secretion into the ECM. When Tango1L is deleted, procollagen accumulates in the ER, activating ER stress pathways and triggering cell death (Clark and Link, 2021). Although only subtle changes are seen when tango1S is mutated, the cartilage ECM is most disrupted when double isoform mutations are present. As this is not explained by higher levels of collagen secretion in tango1L+S fish, it suggests that the levels of ER stress in these fish are highest, which could result in more activation of the unfolded protein response (UPR) (Schröder and Kaufman, 2005). It has previously been shown that activation of the UPR leads to suppression of protein synthesis (Harding et al, 1999; Martínez and Chrispeels, 2003; Pakula et al.,

2003) and upregulation of ER-associated degradation (ERAD) (Casagrande et al., 2000; Friedlander et al., 2000; Travers et al., 2000), which could explain why type II collagen is not seen trapped in every chondrocyte in tango1L+S fish.

Although our results suggest that mutation of Tango1S has a limited effect on collagen secretion into the cartilage ECM by chondrocytes, significant changes were seen to the content of other ECM components including proteoglycans. The interaction between proteoglycans and collagen in the cartilage ECM is vital for collagen fibrillogenesis and structural organisation of the cartilage (Stanescu, 1990; Roughley and Lee, 1994). During development and in diseases such as osteoarthritis, changes to ECM composition and structure such as reduced proteoglycan content influence chondrocyte behaviour and biomechanical force distribution leading to morphological changes to the cartilage (Korhonen et al., 2011; Prein et al., 2016; Danalache et al., 2019), which could potentially predispose Tango1S mutants to earlier onset of ageing joint phenotypes.

Although subtle, the defects seen in ECM secretion in larval zebrafish with a tango1S mutation could lead to cumulative defects with age, such as the changes seen in the spine of adult tango1S mutants. The spinal phenotypes observed in adult tango1S mutants (straighter spine, vertebral misalignments, intervertebral disc degeneration) appear to develop only in adulthood, rather than being a defect in patterning. Minor changes to ECM components such as type X collagen and proteoglycans could lead to altered loading and therefore a cumulative effect on spinal morphology over time as, the intervertebral disc relies heavily on a well-organised ECM for its mechanical properties (Kague et al., 2021) and structural integrity. The observed changes to the discs in tango1S mutant zebrafish are reminiscent of disc degeneration [fibrotic tissue, disorganised nucleus pulposus cells, bulging annulus fibrosus (Kague et al., 2021)] and suggest a breakdown in the integrity of this matrix. While the impact of a tango1S mutation is subtle in larval zebrafish, the adult phenotype suggests that mutation of tango1S eventually leads to more severe changes, particularly in tissues with high mechanical load.

## MATERIALS AND METHODS
### Zebrafish husbandry
Zebrafish were maintained as described in Westerfield (2000). All experiments were approved by the local ethics committee and performed under a UK Home Office Project Licence. ARRIVE guidelines were adhered to, and the completed checklist is submitted as supplementary information.

### CRISPR line generation
Clustered regularly interspace short palindromic repeats (CRISPR) guides against zebrafish tango1 were designed (Table S1, Fig. S1) and ordered as 2 nmol Alt-R™ CRISPR Cas9 crRNA from IDT. This crRNA was annealed to Alt-R® CRISPR-Cas9 tracrRNA, 5 nmol (IDT) at 95°C for 5 min to form a 57 μM gRNA. This gRNA was then incubated with Alt-R™ S.p. Cas9 Nuclease V3 (IDT) (diluted to 57 μM) for 5 min at 37°C to assemble the ribonucleoprotein (RNP) for injection. The yolk of one-cell-stage zebrafish embryos was injected with 1 nl of RNP (1000 pg gRNA and 4700 pg of Cas9), and surviving embryos were raised under normal husbandry conditions. Fish that survived to adulthood were outcrossed to wild-type fish, and the offspring from this cross were grown to adulthood for fin clipping and genotyping to identify fish carrying the same deletions via sequencing.

### DNA extraction and genotyping
Whole larvae or larval fin clips were incubated in 25 mM NaOH, 0.2 mM EDTA at 95°C for 30 min prior to addition of 40 mM Tris HCl, pH 5. To determine the presence of mutations in the TANGO1 isoforms, PCR was

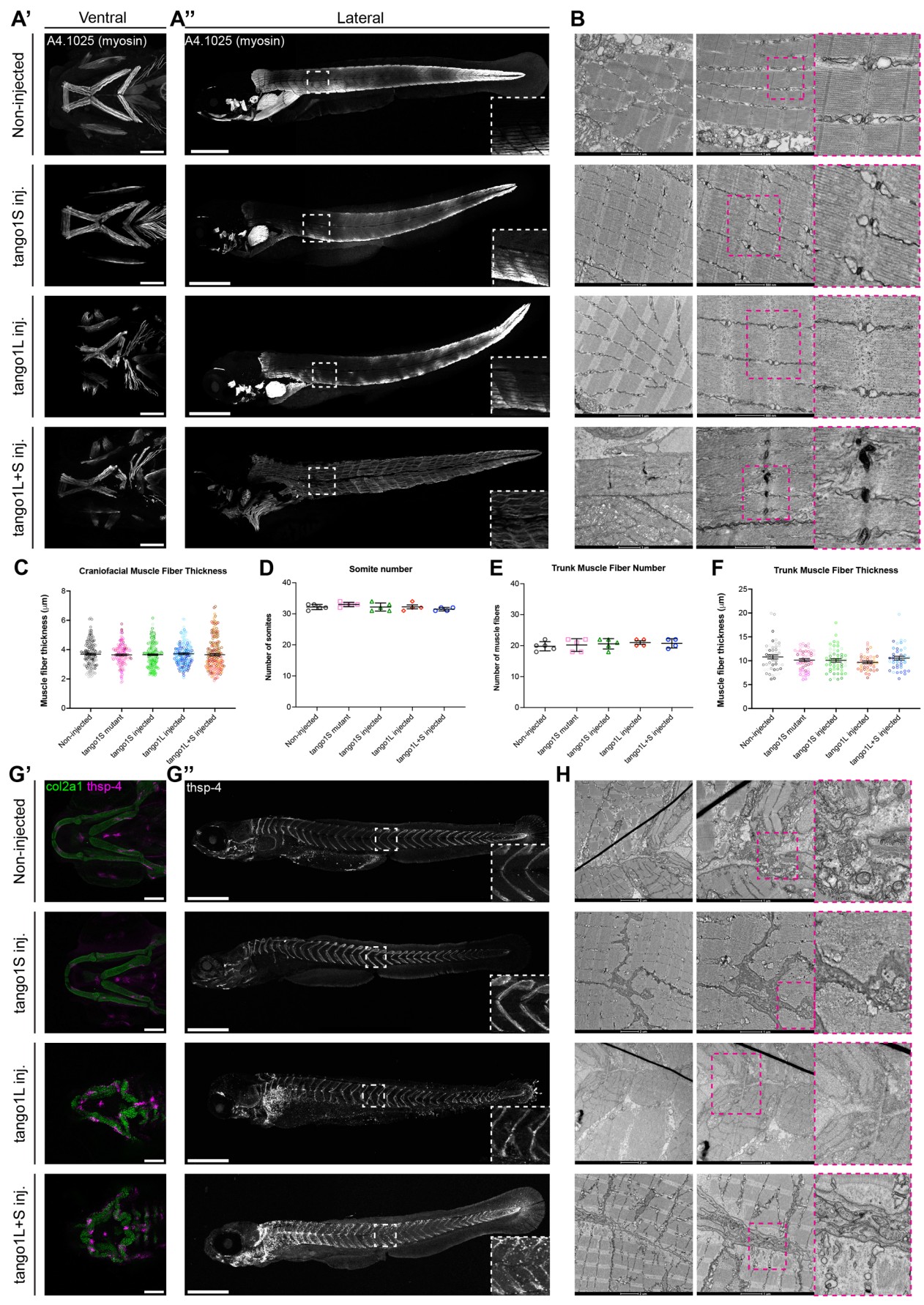

**Fig. 5.** See next page for legend.

**Fig. 5. Muscle packing and trunk muscle attachment sites are disrupted when Tango1 is lost, with the phenotype exacerbated when both isoforms of Tango1 are deleted.** (A′,A″) Maximum projections of ventral (A′) and lateral (A″) view confocal image stacks from 5 dpf larvae stained for myosin. (B) Electron micrographs of 5 dpf zebrafish trunk muscle fibres. (C-F) Quantification of Muscle fibre thickness in the lower jaw (C) and trunk (D), number of muscle fibres per somite (E), and somite number (F) from confocal image stacks. (G′,G″) Maximum projections of ventral (G′) and lateral (G″) view confocal image stacks from 5 dpf larvae stained for thrombospondin-4. Lateral images in A and G are made up of three separate maximum projections stitched together using the Pairwise Stitching plug-in for ImageJ. Scale bars: 100 µm in A′ and G′, and 500 µm in A″ and G″. (H) Electron micrographs of 5 dpf zebrafish myosepta. In B and H, location of higher-resolution electron microscopy images is shown by pink dashed line boxes in left-hand overview image. Kruskal–Wallis test with Dunn's multiple comparison test performed in C and D, one-way ANOVA with Dunn's multiple comparison test performed in E and F. Data are mean with s.e.m.

performed as in Clark and Link (2021). To sequence the region of the mutation, the forward primer was used and samples were Sanger sequenced (Source Bioscience).

### Whole-fish measurements

Fixed larvae were mounted in 70% glycerol and imaged on a Leica MZ10F stereo microscope at 1.8× zoom. The length of the larvae and the swim bladder were measured using the line tool and measure command in ImageJ. To calculate the volume of the swim bladder, the formula $V=4/3\,\pi\,ab^2$ was used (Lindsey et al., 2010), where a=the measurement taken at the longest part of the swim bladder in the horizontal axis and b=the longest measurement in the vertical axis.

### Histological staining and slide imaging

Following euthanasia, fish were fixed in 4% paraformaldehyde (PFA) for 2 h at room temperature, embedded in paraffin and sectioned at 5 µm thickness. The resulting sections were stained with Haematoxylin and Eosin (H&E), Masson's trichrome, Picrosirius Red or Safranin O/Fast Green as described in Lawrence et al. (2021). Stained sections were imaged on an Evident (Olympus) VS200 Slide Scanner at 20× magnification.

### Wholemount immunohistochemistry

Zebrafish larvae were euthanised in tricaine methanesulphonate (MS222) prior to fixation in 4% PFA for 2 h at room temperature and stored in 100% MeOH at −20°C. Larvae were sequentially rehydrated, washed in 0.1% Triton X-100 in 1× phosphate buffered saline (PBST), blocked in 5% horse serum in PBST for 4 h and incubated in primary antibody solution [anti-COL2A1 (Developmental Studies Hybridoma Bank, II-II6B3, mouse, 1:50); anti-THBS4 (Abcam, ab211143, rabbit, 1:500); anti-GFP (Abcam, ab13970, chicken, 1:500); anti-sox9a (Abcam, ab209820, rabbit, 1:500); anti-col10a1 (Developmental Studies Hybridoma Bank, mouse X-AC9, 1:50); anti-col1 (Abcam, ab23730, rabbit, 1:100); anti-myosin heavy chain (Developmental Studies Hybridoma Bank, A4.1025, mouse, 1:100)] overnight at 4°C with gentle agitation. Following this, larvae were washed in PBST, re-blocked for 2 h and incubated in secondary antibody solution (AlexaFluor donkey anti-mouse 488, AlexaFluor donkey anti-rabbit 568, both 1:500) and/or WGA conjugate (CF®640R WGA Biotium, 2 µg/ml) overnight at 4°C with gentle agitation. Finally, larvae were incubated in 5 µg/ml 4′,6-diamidino-2-phenylindole (DAPI) for 30 min prior to imaging.

### Confocal imaging

Stained zebrafish larvae were mounted on a coverslip in 0.5% low-melting-point (LMP) agarose and imaged on a Leica SP5II or SP8 confocal microscopes with a 10× objective. For live larval imaging of the *krt19:col1a2-GFP* (Morris et al., 2018), *col2a1aBAC:mCherry* (Hammond and Schulte-Merker, 2009) and *Ola.Sp7:NLS-GFP* (Spoorendonk et al., 2008) transgenic lines, larvae were anaesthetised in 0.1 mg/ml MS222 and mounted on a coverslip in warm 0.3% LMP agarose with MS222 prior to imaging on a Leica SP5II confocal microscope with a 63× objective.

### Jaw and joint measurements

The width and length of the lower jaw, along with the jaw joint width, and Meckel's cartilage depth were measured from confocal image stacks of type II collagen immunostained zebrafish larvae using the line tool and measure command in ImageJ.

### Visualisation and quantification of jaw movement

Jaw movement was recorded and analysed as in Schilling and Le Pabic (2009) and Lawrence et al. (2018). Briefly, anaesthetised larvae were mounted in LMP agarose, and the agarose around the head was removed to allow free movement of the jaw recordings made at 1 ms intervals over 1000 frames on an Olympus IX73 microscope (10× lens) using the Micro-Manager 1.4.16 plugin for ImageJ (Edelstein et al., 2014). The number of jaw movements was counted manually, and the maximum displacement was measured using the line tool in ImageJ.

### Wholemount alcian blue stain

Staining of larvae with Alcian Blue [0.02% Alcian Blue (Sigma-Aldrich) in 70% EtOH with 80 mM MgCl₂] was performed as described in Walker and Kimmel (2007) on larvae fixed in 3.5% formaldehyde. Stained larvae were imaged on a Leica MZ10F stereo microscope.

### Quantification of staining intensity

The staining intensity from confocal image stacks and brightfield images was performed in ImageJ (Schindelin et al., 2012). The segmented line tool was used to draw a line through the ECM surrounding chondrocytes, and the plot profile command performed to extract the grey value along this line. This was performed on multiple sections from three fish per condition, and measurements were normalised to the image background.

### Electron microscopy

Whole larvae were fixed in 2.5% glutaraldehyde in 0.1 M sodium cacodylate for 1 h at room temperature under agitation. Samples were then washed in 0.1 M sodium cacodylate and processed for TEM. Fixed samples were embedded in 3% agarose before being osmium/uranyl acetate stained, dehydrated and infiltrated with EPON in a Leica EM TP tissue processor using the standard protocol outlined in Lawrence (2021). Ultra-thin sections were cut at a thickness of 70 nm and stained with uranyl acetate before being imaged on a Thermo Fisher Scientific Talos L120C 120 kV transmission electron microscope.

### µCT

Adult zebrafish (14 months) were fixed in 4% PFA for 1 week and sequentially dehydrated in 70% ethanol. Full body scans were taken with a voxel size of 20 µm using a Nikon XT H 225ST µCT scanner (x-ray source of 130 kV and 53 µA without additional filters). The generated scans were then reconstructed using Nikon CT Pro 3D software. During reconstruction, greyscale values were calibrated against a scan of a phantom with known density (0.75 g/cm³). Volume rendering was performed using Amira 6.0 software (Thermo Fisher Scientific).

### Quantification of spine curvature

Snapshot images of µCT scans where the fish was in a lateral orientation were loaded into ImageJ, and the average curvature of the whole spine, first 14 vertebrae (referred to as the thoracic spine) and last 14 vertebrae (referred to as the lumbar spine) was quantified using the Kappa plugin.

### Power calculations

Effect size and standard deviation were estimated from previous experiments, and power calculations were performed determine the appropriate sample size to reject or accept the null hypothesis. These calculations were performed in PS: Power and Sample Size Calculation software [version 3.1.6, (Dupont and Plummer, 1990)].

### Statistics

All statistical analyses were performed in GraphPad Prism (version 10.2.0), and the null hypothesis was rejected at a $P$-value of 0.05 or lower. For all datasets, D'Agostino and Pearson tests for normality were performed followed by the appropriate statistical test (detailed in figure legends).

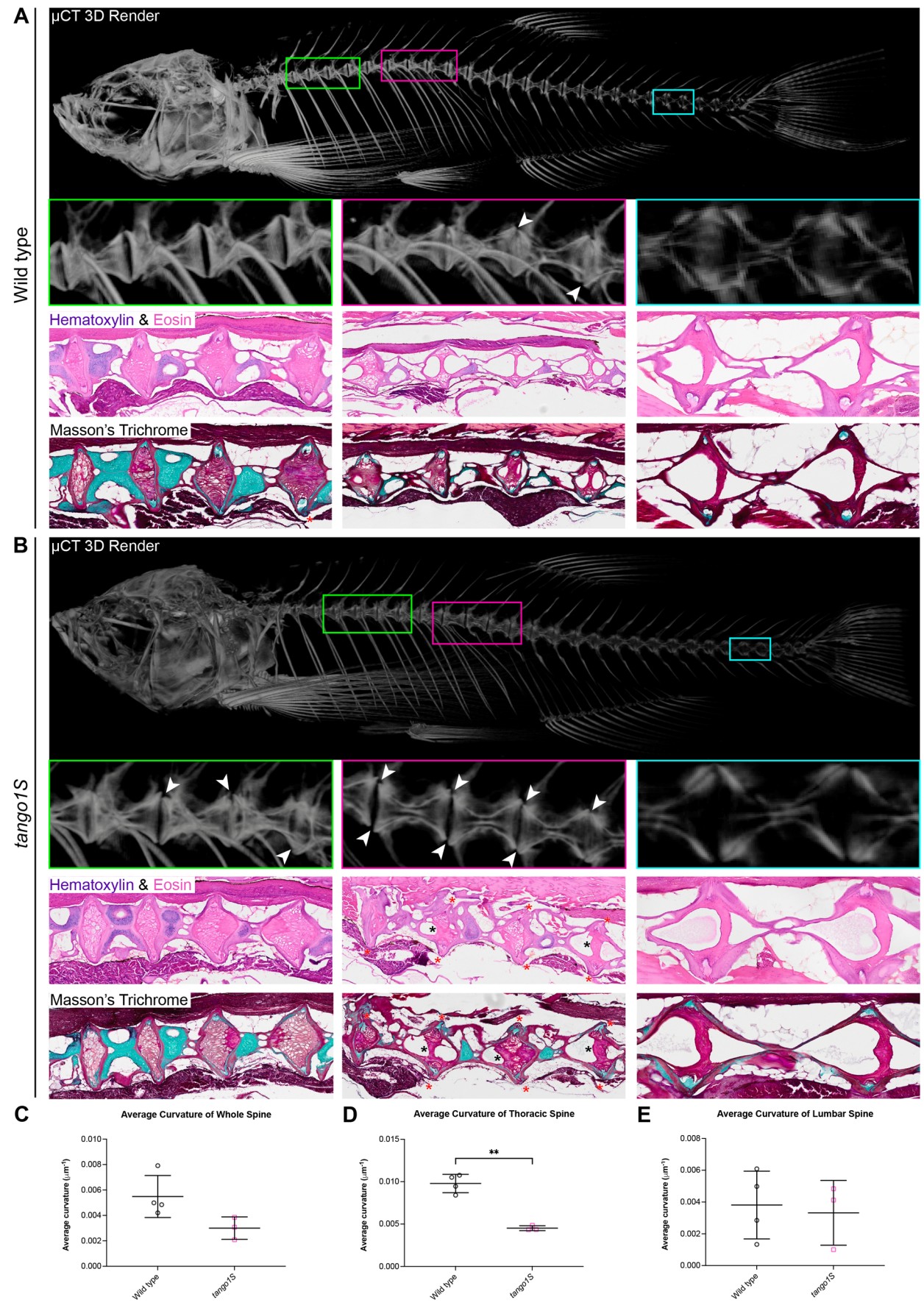

**Fig. 6.** See next page for legend.

**Fig. 6. Adult zebrafish with a Tango1S mutation have a subtle spine phenotype characterised by a reduction in spinal curvature, vertebral misalignments and changes to the intervertebral discs.** (A,B) Images from a representative wild-type (A) and Tango1S mutant (B) zebrafish at 14 months of age. Images include a three-dimensional render from µCT scans, with regions of interest highlighted by coloured boxes. For each region of interest, the three-dimensional render is shown with the corresponding histological sections stained with H&E or Masson's trichrome shown below. White arrowheads=vertebral misalignments, black asterisks=areas of fibrotic tissue and cellular disorganisation in the NP, red asterisks=areas of annulus fibrosus stretching. (C-E) Quantification of whole (C), thoracic (D), and lumbar (E) spine curvature in adult fish from µCT scans. Shapiro–Wilk test for normality followed by Mann–Whitney $t$-test performed in C and Welch's $t$-test in D and E. Data are mean with s.e.m. **$P \leq 0.01$.

## Acknowledgements

We would like to thank the Animal Service Unit staff and personnel from the Wolfson Bioimaging facility for their help in this project. We are saddened to report that Professor David Stephens died in 2024, prior to the submission of this manuscript. As authors, we wish to pay tribute to him; he was an excellent scientist, mentor and friend, and we send our continued good wishes to his family.

## Competing interests

The authors declare no competing or financial interests.

## Author contributions

Conceptualization: E.A.L., B.L., D.J.S.; Data curation: E.A.L., D.J.S., C.L.H.; Formal analysis: E.A.L., Q.T., B.F., R.M.C.; Funding acquisition: B.L., D.J.S., C.L.H.; Investigation: E.A.L., M.E.P.-S., Q.T., B.F.; Methodology: E.A.L., R.M.C., M.D., B.L., C.L.H.; Project administration: E.A.L., D.J.S., C.L.H.; Resources: D.J.S., C.L.H.; Supervision: D.J.S., C.L.H.; Validation: E.A.L.; Visualization: E.A.L., Q.T.; Writing – original draft: E.A.L., B.L., C.L.H.; Writing – review & editing: E.A.L., B.F., R.M.C., C.L.H.

## Funding

This research was supported by a grant from the Biotechnology and Biological Sciences Research Council (BBSRC; BB/V004352/1). Open Access funding provided by University of Bristol. Deposited in PMC for immediate release.

## Data and resource availability

Raw data are available at the University of Bristol data repository, data.bris, at https://doi.org/10.5523/bris.26hh7hyxs77nr2u07eriovgpu4. All other relevant data and details of resources can be found within the article and its supplementary information. The Tango 1S stable fish mutant line is available on request (and subsequently will be available from the European Zebrafish Resource Center).

## Peer review history

The peer review history is available online at https://journals.biologists.com/bio/lookup/doi/10.1242/bio.062117.reviewer-comments.pdf

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
