## [Peer Review File · Biology Open]

The short isoform of Tango1 is dispensable for zebrafish survival but is required for skeletal patterning and integrity.

Elizabeth A. Lawrence, Maria Esther Prada-Sanchez, Qiao Tong, Bianca Fernandes, Rebecca M. Chatwin, Michael Donohue, Brian A. Link, David J. Stephens and Chrissy L. Hammond

DOI: 10.1242/bio.062117

Editor: Tristan Rodríguez

Review timeline

Original submission:	5 July 2025
Editorial decision:	14 July 2025
First revision received:	20 October 2025
Accepted:	22 October 2025

Original submission

First decision letter

MS ID#: bio.062117

MS Title: The short isoform of Tango1 is dispensable for zebrafish survival but is required for skeletal patterning and integrity.

Authors: Elizabeth A Lawrence; Maria Esther Prada-Sanchez; Qiao Tong; Bianca Fernandes; Rebecca M Chatwin; Michael Donohue; Brian A Link; David J Stephens; Chrissy L Hammond
Article Type: Research Article

I have now reached a decision on the above manuscript.

The reviewer reports are shown at the bottom of this email or can be accessed, together with a copy of this decision letter, by going to:

As you will see, the reviewers raised a number of substantial criticisms that prevent me from accepting the paper at this stage.

They suggest, however, that a revised version might prove acceptable, if you can address their concerns. If you think that you can deal satisfactorily with the criticisms on revision, I would be pleased to see a revised manuscript. We would then return it to the reviewers.

At this stage, we also ask you to ensure your manuscript complies with our formatting guidelines. Provided you are able to fully address the referees' comments, we are positive about publication of your paper (we accept over 95% of revision submissions) and therefore hope you won't mind any extra work involved in reformatting your manuscript at this point.

Please ensure that you clearly highlight all changes made in the revised manuscript. Please avoid using 'Tracked changes' in Word files as these are lost in PDF conversion.

I should be grateful if you would also provide a point-by-point response detailing how you have dealt with the points raised by the reviewers in the 'Response to Reviewers' box. Please attend to

all of the reviewers' comments. If you do not agree with any of their criticisms or suggestions please explain clearly why this is so.

Reviewer 1

Comments for the author

In this paper, Lawrence and colleagues use zebrafish to investigate the consequences of knocking out the long and short isoforms of TANGO1, a protein crucial for the export of large cargo from the endoplasmic reticulum (ER). Previous work showed that the long isoform of tango1 (tango1L) is essential for proper secretion of collagen and other large ECM components, but the role of the short isoform (tango1S) remained completely unexplored. To tackle this, the authors use CRISPR/Cas9 to knockout tango1L, tango1S or both in zebrafish and investigate their phenotypes. Because tango1L and tango1L+tango1S crispants do not survive past 12 dpi, but tango1S crispant do, the authors conclude that the short isoform of tango1 can at least partly compensate for the loss of the long isoform. The authors then delve deeper into this, first by looking more closely into the lower jaw, known to be severely affected in tango1L mutants. Using a range of histological stainings and transgenic zebrafish, the authors find that tango1S seems to indeed have a partially compensatory role in ECM secretion of both fibrillar and non-fibrillar collagens (clearly visualised as puncta inside chondrocytes) as well as carbohydrate components of the ECM. In line with this, the most severe secretion phenotypes are observed in tango1L+tango1S crispants. Importantly, the authors extend their phenotypic characterisation of crispants to the skin and muscle, a tissue with lower collagen content. In these tissues too, loss-of-function of tango1 isoforms disrupt the ECM and tissue organisation to a larger (tango1L+tango1S and tango1L) or smaller (tango1S) degree. Because tango1S crispants live to adulthood, the authors generate tango1S mutants to study the long-term consequences of the functional loss of tango1S. This led them to some interesting observations in the vertebral column of tango1S mutants, including fibrosis in the nucleus pulposus and changes in the curvature of the spine, reminiscent of intervertebral disc degeneration and other pathologies seen in patients in TANGO1 mutations.

Overall, this is a solid paper. The authors use the right methods to address their questions and often use two independent methods to answer the same question, which adds robustness to their study. Part of the results presented in this paper (related to tango1L) align with those from Clark and Link 2021, confirming the reproducibility of the findings. The author's conclusions are well supported by the data and as such, no more experiments are necessary to complete their story. References to the literature are appropriate and comprehensive and the authors clearly refer to other recent studies on TANGO1 and put them in context with their own work.

I only have some minor comments and suggestions to improve the paper prior to publication in Biology Open:

1. The authors chose to use non-injected controls as their controls. Perhaps injecting embryos with Cas9 but no guides or non-targeting guides would have been a better control. While I don't think it is necessary to add such controls at this point, can the authors explain why they chose non-injected embryos as controls?
2. All figures: Please indicate what post-hoc statistical test was used to correct for multiple comparisons (e.g. Tukey, Bonferroni, Dunnett's, etc.)
3. Figure 1A. Please indicate in the figure legend how many zebrafish were tracked for each experimental group of in the survival plot.
4. Materials and Methods. Please indicate for how long zebrafish were fixed in 4% PFA.
5. The first two paragraphs of the Discussion are a short review of the literature and seem redundant with the Introduction. Are they really necessary? In my view, the Discussion could be improved by focusing on explaining the meaning of the results of the paper straightaway and then comparing them to previous studies, discussing their implications and future research directions (as the authors do).

Reviewer 2

Comments for the author

The manuscript by Lawrence et al. is a mainly descriptive work, that addresses the consequences of the loss of function of tango1 in zebrafish development. Previous work has already studied the role of tango1 in zebrafish (Clark and Link, Mol Cell Biol 2021) or medaka (Yasuda et al, Cell Struct Function 2025), and although an important part of the results shown here are confirmatory, this work also provides interesting new evidence regarding possible specific roles for the short isoform of the gene (tango1S). The phenotypic characterization of crispants for both long and short isoforms, as well as the complete tango1 crispant, and also in some instances of the tango1S mutant line, are of high quality and complement previous publications. The description of the phenotype of tango1S loss of function is a valuable addition to the field.

However, addressing certain issues of the manuscript would help to improve it, helping to make a stronger point regarding its main conclusions. Some of these issues are the following:

1. A much better description of Tango1 isoforms should be provided in the Introduction. There is hardly any reference to them, in which organisms have they been characterized, and what is the available information about their function.
2. A clear diagram depicting the structure of the tango1 gene in zebrafish should be included, as supplementary figure 1 is hardly of any use. It should also include information relating to the targeting strategy used to generate crispants and the expected outcomes.
3. Evidence must be provided for the correct targeting and genotypes of the various alleles. The exact editing in the mutant tango1S line should also be shown.
4. Evidence regarding the outcome on gene expression for each of the different crispants should also be provided, to prove that each isoform is correctly targeted. This could be done by RT-qPCR and/or Western Blot (best both).
5. Fig. 1I, J is mislabeled, it should be "Meckel's cartilage depth" and "Meckel's cartilage joint width".
6. lines149-150: how does lack of changes to the digestive system imply that "altered jaw development prevents the fish from feeding normally"? This should be better explained or removed.
7. Fig 6: can the straighter spine phenotype of tango1S mutants be quantified in some way?
8. The Discussion should be considerably shortened. Repeating details of the Results section should be avoided, and the authors should go straight to the point of what can be concluded from the data. In this regard, the authors must try to make a stronger argument on what can be considered phenotypes and functions directly related to the tango1S isoform.

Reviewer's Responses to Questions

Experimental quality

Does each figure have the proper controls?

If 'No', please indicate reasons in Comments for Author box below.

Reviewer #1:

- Yes

Reviewer #2:

- Yes

Were the data analyzed using appropriate statistical tests?
If 'No', please indicate reasons in Comments for Author box below.

Reviewer #1:

- Yes

Reviewer #2:

- Yes

Reproducibility

Were experiments performed using adequate number of biological replicates?
If 'No', please indicate reasons in Comments for Author box below.

Reviewer #1:

- Yes

Reviewer #2:

- Yes

Does the methods section provide sufficient detail to permit reproducibility?
If 'No', please indicate reasons in Comments for Author box below.

Reviewer #1:

- Yes

Reviewer #2:

- No

Completeness

Are the manuscript's conclusions supported by the data?
If 'No', please indicate reasons in Comments for Author box below.

Reviewer #1:

- Yes

Reviewer #2:

- Yes

Scholarship

Do the authors cite and discuss the merits of data that would argue for and against their conclusion?

If 'No', please indicate reasons in Comments for Author box below.

Reviewer #1:

- Yes

Reviewer #2:

- Yes

Does the manuscript title & abstract accurately reflect the contents of the manuscript, without hyperbole?

If 'No', please indicate reasons in Comments for Author box below.

Reviewer #1:

- Yes

Reviewer #2:

- Yes

First revision

Author response to reviewers' comments

Reviewer 1: *In this paper, Lawrence and colleagues use zebrafish to investigate the consequences of knocking out the long and short isoforms of TANGO1, a protein crucial for the export of large cargo from the endoplasmic reticulum (ER). Previous work showed that the long isoform of tango1 (tango1L) is essential for proper secretion of collagen and other large ECM components, but the role of the short isoform (tango1S) remained completely unexplored. To tackle this, the authors use CRISPR/Cas9 to knockout tango1L, tango1S or both in zebrafish and investigate their phenotypes. Because tango1L and tango1L+tango1S crispants do not survive past 12 dpi, but tango1S crispant do, the authors conclude that the short isoform of tango1 can at least partly compensate for the loss of the long isoform. The authors then delve deeper into this, first by looking more closely into the lower jaw, known to be severely affected in tango1L mutants. Using a range of histological stainings and transgenic zebrafish, the authors find that tango1S seems to indeed have a partially compensatory role in ECM secretion of both fibrillar and non-fibrillar collagens (clearly visualised as puncta inside chondrocytes) as well as carbohydrate components of the ECM. In line with this, the most severe secretion phenotypes are observed in tango1L+tango1S crispants. Importantly, the authors extend their phenotypic characterisation of crispants to the skin and muscle, a tissue with lower collagen content. In these tissues too, loss-of-function of tango1 isoforms disrupt the ECM and tissue organisation to a larger (tango1L+tango1S and tango1L) or smaller (tango1S) degree. Because tango1S crispants live to adulthood, the authors generate tango1S mutants to study the long-term consequences of the functional loss of tango1S. This led them to some interesting observations in the vertebral column of tango1S mutants, including fibrosis in the nucleus pulposus and changes in the curvature of the spine, reminiscent of intervertebral disc degeneration and other pathologies seen in patients in TANGO1 mutations.*

Overall, this is a solid paper. The authors use the right methods to address their questions and often use two independent methods to answer the same question, which adds robustness to their study. Part of the results presented in this paper (related to tango1L) align with those from Clark and Link 2021, confirming the reproducibility of the findings. The author's conclusions are well supported by the data and as such, no more experiments are necessary to complete their story. References to the literature are appropriate and comprehensive and the authors clearly refer to other recent studies on TANGO1 and put them in context with their own work.

I only have some minor comments and suggestions to improve the paper prior to publication in Biology Open:

We thank the reviewer for their time and for providing us with a positive response and constructive feedback. We value their suggestions and have responded to each one with the actions taken shown in green under the corresponding comment.

Changes to the original manuscript are highlighted in yellow.

1. The authors chose to use non-injected controls as their controls. Perhaps injecting embryos with Cas9 but no guides or non-targeting guides would have been a better control. While I don't think it is necessary to add such controls at this point, can the authors explain why they chose non-injected embryos as controls?

Some of the experiments were completed with a sham injection condition and no difference was seen to non-injected fish when looking at survival and general development, so in the interest of preserving valuable reagents (Cas9) and reducing the number of embryos subjected to the injection

process, we proceeded with non-injected fish as the control group. This decision has been addressed in the results section, lines 150-154.

2. All figures: Please indicate what post-hoc statistical test was used to correct for multiple comparisons (e.g. Tukey, Bonferroni, Dunnett's, etc.)

Details of the specific post-hoc statistical test used to correct for multiple comparisons has been listed and highlighted in the figure legends.

3. Figure 1A. Please indicate in the figure legend how many zebrafish were tracked for each experimental group of in the survival plot.

The number of zebrafish tracked for each group in the survival experiment has been added and highlighted to the figure legend at line 849.

4. Materials and Methods. Please indicate for how long zebrafish were fixed in 4% PFA.

The duration of PFA fix has been added and highlighted in the methods section at lines 475 and 483.

5. The first two paragraphs of the Discussion are a short review of the literature and seem redundant with the Introduction. Are they really necessary? In my view, the Discussion could be improved by focusing on explaining the meaning of the results of the paper straightaway and then comparing them to previous studies, discussing their implications and future research directions (as the authors do).

To address this point, some of the details in these paragraphs have been moved to add detail to the introduction lines and the redundant sections removed to make the discussion more focused.

Reviewer 2: *The manuscript by Lawrence et al. is a mainly descriptive work, that addresses the consequences of the loss of function of tango1 in zebrafish development. Previous work has already studied the role of tango1 in zebrafish (Clark and Link, Mol Cell Biol 2021) or medaka (Yasuda et al, Cell Struct Function 2025), and although an important part of the results shown here are confirmatory, this work also provides interesting new evidence regarding possible specific roles for the short isoform of the gene (tango1S). The phenotypic characterization of crispants for both long and short isoforms, as well as the complete tango1 crispant, and also in some instances of the tango1S mutant line, are of high quality and complement previous publications. The description of the phenotype of tango1S loss of function is a valuable addition to the field.*

However, addressing certain issues of the manuscript would help to improve it, helping to make a stronger point regarding its main conclusions. Some of these issues are the following:

We thank the reviewer for their time and for providing us with constructive feedback. We value their suggestions and have responded to each one with the actions taken shown in green under the corresponding comment. Changes to the original manuscript are highlighted in yellow.

1. A much better description of Tango1 isoforms should be provided in the Introduction. There is hardly any reference to them, in which organisms have they been characterized, and what is the available information about their function.

Detail regarding the specific isoforms and their domains has been added to paragraph 3 of the introduction to address this. Further information into studies into each of the isoforms has also been added to paragraph 4 of the introduction.

2. A clear diagram depicting the structure of the tango1 gene in zebrafish should be included, as supplementary figure 1 is hardly of any use. It should also include information relating to the targeting strategy used to generate crispants and the expected outcomes.

We have generated a new Supp Figure 1 which we hope is more useful.

3. Evidence must be provided for the correct targeting and genotypes of the various alleles. The exact editing in the mutant tango1S line should also be shown.

We have generated a new Supp Figure 1 which we hope is more useful.

4. Evidence regarding the outcome on gene expression for each of the different crispants should also be provided, to prove that each isoform is correctly targeted. This could be done by RT-qPCR and/or Western Blot (best both).

Unfortunately we haven't been able to identify antibodies for zebrafish that work well on Western blots (without lots of background unspecific bands). We have performed RT-qPCR (and spent considerable time optimising) however, we do not see a significant difference to expression levels of either the Tango L or S isoforms. While qPCR can sometimes be useful if nonsense mediated decay does lead to specific degradation of the transcript this is not always the case, and all possibilities (increased, decreased and no change) can be observed for different genes dependent on the feedback loops operating. We have given detail of the methodology used and the data below. But as it is a null result we haven't included it in the paper. We feel that having the sequence data and the specific phenotypes does demonstrate the correct targeting of the isoforms and that together with the updates to Sup Fig 1, we hope that this addresses the issue sufficiently.

5. Fig. 1I, J is mislabeled, it should be "Meckel's cartilage depth" and "Meckel's cartilage joint width".

Thanks for picking this up. The two graphs in Figure 1I and J have been re-labelled to 'Meckel's Cartilage Depth' and 'Meckel's Joint Width' respectively.

6. lines 149-150: how does lack of changes to the digestive system imply that "altered jaw development prevents the fish from feeding normally"? This should be better explained or removed.

The statement "altered jaw development prevents the fish from feeding normally" has been removed.

7. Fig 6: can the straighter spine phenotype of tango1S mutants be quantified in some way?

The curvature of the spine has now been quantified using the Kappa plugin for ImageJ. Details of this analysis have been added to the methods section at line 547-551 and the results added including graphs of the analysis included in an updated figure 6 (Fig 6D).

8. The Discussion should be considerably shortened. Repeating details of the Results section should be avoided, and the authors should go straight to the point of what can be concluded from the data. In this regard, the authors must try to make a stronger argument on what can be considered phenotypes and functions directly related to the tango1S isoform.

Redundant information which repeats the introduction has been removed and the summary of results included in the discussion dramatically shortened.

Extra information on re: gene expression - in response to point 4 from reviewer 2.

Methods used for gene expression analysis by RT-qPCR.

Gene expression analysis by RT-qPCR

Gene expression levels of *Mia3S* and *Mia3L* were assessed by real-time PCR in zebrafish larvae at 5 dpf. We utilized 10 larvae per pool and performed 3 distinct pools per group. The endogenous gene *bActin* was used as a normalization control.

RNA was extracted following the Trizol method and stored at -80°C after elution in Milli-Q water. Total RNA was quantified with a NanoDrop 2000 (thermo scientific), and cDNA was synthesized

from 1000 ng of total RNA using the Invitrogen™ SuperScript™ IV First-Strand Synthesis System (Thermo Fisher Scientific, Cat. No. 18091050), according to the manufacturer's instructions.

cDNA samples underwent PCR using SYBR™ Green PCR Master Mix (Applied Biosystems, Cat. No. A25742) and were amplified in an QuantStudio 3 Real-Time PCR System (Applied biosystems by thermofisher). For *Mia3S*, annealing was performed at 60°C, and for *Mia3L* at 62°C. Gene expression was normalized against wild-type controls and the least variable endogenous gene.

Relative quantification was calculated using the $2^{-\Delta\Delta CT}$ method by Livak and Schmittgen (2001) (doi: 10.1006/meth.2001.1262), where ΔCT is the difference in expression between the target and endogenous control genes, and $\Delta\Delta CT$ is the ΔCT difference between the mutant and wild-type samples for the same developmental period. Each sample was processed in triplicate.

The nucleotide sequences of the primer pairs used are described in Table 2. Primers for *Actin* (doi: 10.1186/1471-2199-9-102) and *Mia3L* (doi: 10.1091/mbc.E20-11-0745) have been previously reported. Primers for amplifying *Mia3S* were:

Mia3L

- Primer 1: AGAGGTCCAGGCGGAAA
- Primer 2: CACACATCGCCCCGTTT

Mia3S

- Primer F: ATCCTGTCCAGACTTCACGAAC
- Primer R: AGCACCGTCCTCCAGAAGAA

Data from the RT-qPCR

We have removed two figures, which were provided for the referees in confidence.

Second decision letter

MS ID#: bio.062117R1

MS TITLE: The short isoform of Tango1 is dispensable for zebrafish survival but is required for skeletal patterning and integrity.

AUTHORS: Elizabeth A Lawrence; Maria Esther Prada-Sanchez; Qiao Tong; Bianca Fernandes; Rebecca M Chatwin; Michael Donohue; Brian A Link; David J Stephens; Chrissy L Hammond

I am happy to tell you that your manuscript has been accepted for publication in Biology Open, pending our standard publication integrity checks. It was accepted on 22nd October 2025.